# Dynamic Optimal Decision Making of Innovative Products' Remanufacturing Supply Chain

**Lang Liu** [1,2]**, Zhenwei Liu** [2,*]**, Yutao Pu** [2] **and Nan Wang** [1]

1   School of Business Administration, Guizhou University of Finance and Economics, Guiyang 550025, China
2   School of Economic and Management, East China Jiaotong University, Nanchang 330013, China
*   Correspondence: 2020048025200008@ecjtu.edu.cn; Tel.: +86-189-4235-0433

**Abstract:** In order to realize the recyclability of innovative product resources, we explored the optimal dynamic path of each decision variable in the remanufacturing supply chain and analyzed the impact of each decision variable on supply chain performance. Based on the Bass innovation diffusion model, we established a remanufacturing supply chain model in which a single manufacturer leads and a single retailer follows, and the retailer is responsible for recycling. The optimal wholesale price, retail price, and recovery effort path were obtained through optimal control theory. We also discussed the influence of different innovation coefficients and imitation coefficients on the overall long-term profit of each member in the supply chain, and at the same time, found the optimal market share of the product. The research results show that the larger the market innovation coefficient and the imitation coefficient are, the larger the overall long-term profit of the manufacturer and the greater the market share of the product, while the overall long-term profit of the retailer and the entire supply chain will increase first and then decrease; when the innovation coefficient and imitation coefficient are above a certain level, retailers will not enter the market. In a market with a small innovation coefficient and a large imitation coefficient, the overall long-term profits of retailers and supply chains will be higher. This study provides a theoretical basis for the decision making of the remanufacturing supply chain of innovative products in a dynamic environment, and also provides guidance for the practice of nodal enterprises in the supply chain.

**Keywords:** innovative products; remanufacturing; supply chain management; differential game; bass model

## 1. Introduction

In recent years, with the rapid development of the economy, the speed of innovation product update iteration is accelerating. While consuming a lot of resources, it also produces a large number of waste products and causes serious environmental pollution. If these waste products are not recycled, it will not only cause environmental pollution, but also leads to a great waste of resources. Therefore, environmental issues have received more and more attention [1], and awareness of environmental issues and social responsibility is increasing [2]. The White Paper 2018 on intelligent remanufacturing products edited by Tsinghua University pointed out that the implementation of waste product recycling and remanufacturing can save 70% of materials, reduce 60% of energy consumption, and reduce 80% of pollutant emissions [3]. At present, governments and enterprises are vigorously promoting waste product recycling and remanufacturing industry to achieve sustainable economic development. European countries have the world's most advanced waste information and communication equipment recycling management systems [4]. The Chinese government has also continuously strengthened the conservation and utilization of resources, strengthened ecological construction and governance, and resolutely taken a new path of green, low-carbon, and environmental protection. Therefore, the remanufacturing supply chain has become an important way to achieve the goals of 'carbon peaking and

carbon neutrality'. Recycling and remanufacturing end-of-life products is a better way for the closed-loop supply chain, which can reduce waste discharge and resource consumption and achieve the purpose of sustainable development [5]. Some large international companies, such as Xerox, Kodak, Hewlett-Packard, Volvo, and BMW, have begun to adopt remanufacturing strategies [6,7]. Many Chinese companies are also following up, such as the recycling and remanufacturing of old mobile phones by Chinese domestic mobile phone manufacturers OPPO and VIVO. Companies adopting remanufacturing strategies are widely used in industries such as computers, printers, ink transportation, furniture, auto parts, and medical equipment [8]. SpaceX's recovery of the Falcon rocket is the pinnacle of the current remanufacturing supply chain. With the vigorous development of the global economy and the widespread use of advanced technology, many diversified products are flooding consumers' lives like an unstoppable wave. The life cycle of products has become shorter and shorter, and the speed of updating products has become faster and faster, making the total amount and growth rate of waste continue to rise. The remanufacturing supply chain is considered one of the most effective ways to deal with waste economically and environmentally, and it has become the consensus of industry and academia on the recycling and reuse of end-of-life products [9,10]. Therefore, it is especially important to analyze the game behavior among the members of the remanufacturing supply chain, coordinate the relationship between them, and maximize the profit of each member.

The remanufacturing supply chain is the supply chain that includes the remanufacturing industry. Remanufacturing refers to the process of product disassembly and recovery in the reverse or closed-loop supply chain; this disassembly and recovery process entails the repair or replacement of outdated product components and/or of expired products [11]. Through the implementation of repair and transformation of these products, their quality and performance are brought back to the standards and requirements of new products. Scholars around the world have conducted a lot of research on the remanufacturing supply chain. Masoudipour et al. [10] first established a reverse-channel design framework from the supply chain perspective and proposed three recycling methods for manufacturers. Jena et al. [12] discussed the impact of whether or not to share advertising costs on second-hand product recycling activities. Tsao et al. [13] proposed a remanufacturing supply chain network adopted by RFID. Bhattacharya et al. [14] explored the problem of second-hand product price optimization in a remanufacturing supply chain system. Ma et al. [15] studied a three-level supply chain consisting of a manufacturer, a retailer, and two recyclers, and considered various cooperation models. Xiong et al. [16] discussed and compared the remanufacturing supply chain in two different situations: manufacturer remanufacturing and retailer remanufacturing. Xie et al. [17] developed a remanufacturing supply chain with online and offline dual-channel sales. They obtained manufacturer–retailer advertising cooperation strategies in two different situations: centralized decision making and decentralized decision making. Zhang et al. [9] analyzed and compared three remanufacturing supply chain models of manufacturer recycling, retailer recycling, and third-party platform recycling from the perspectives of the environment, the economy, and social welfare. Zhang et al. [18] explored the pricing and collection strategies of different closed-loop supply chain models in regard to a quality and marketing effort-dependent demand in a fuzzy environment. He et al. [19] proposed a closed-loop supply chain model consisting of one manufacturer and one third-party collector to investigate competitive collection and channel convenience.

The above studies have greatly enriched the theoretical results of the remanufacturing supply chain, but they are all static models. They do not take into account the dynamic characteristics of the recycling process. With the advent of the information age and the popularity of the Internet, information transmission is becoming more and more rapid, the response of the supply chain to market changes is becoming more and more rapid, and the decision of each member of the supply chain will be able to change in real time; in this circumstance, the production, sales, recycling, and remanufacturing between manufacturers and retailers is a long-term differential game process, so the static decision-making assump-

tion will no longer apply. Lee [20] first proposed and studied the remanufacturing supply chain model under decentralized decision making with dynamic features. In the dynamic remanufacturing supply chain model, De Giovanni [21] considered a remanufacturing supply chain composed of a manufacturer and a retailer investing in green advertising based on the goodwill model. Xiang and Xu [22] researched a remanufacturing supply chain consisting of a manufacturer, a retailer, and an Internet service platform that invests in research and development advertising and extensive data marketing. He et al. [23] studied a low-carbon service supply chain composed of a service provider and a service integrator and found that a two-way contract can benefit the entire service supply chain and its members. Song et al. [24] constructed a remanufacturing supply chain model of shared bicycles that considers the impact of recycling efforts on the riding experience. Yang and Xu [25] developed a remanufacturing supply chain model with multiple manufacturing and remanufacturing plants and multiple recycling and supply centers in a low-carbon context and derived the optimal decisions of supply chain participants.

The above studies have discussed dynamic remanufacturing supply chains in various situations. Still, all assume that the product market is relatively mature, and use classic demand models or goodwill models to describe changes in demand. However, there is a gradual promotion process for a new product entering the market. Few studies describe this process in remanufacturing supply chain coordination. According to the China Household Electrical Appliance Research Institute 2022 report data, 2021 China's computer population comprised 320 million units, 1.5 billion mobile phone units, 2421.8 million units of theoretical scrap of computers, and more than 400 million units of theoretical scrap of mobile phones [26]. The remanufacturing of these used innovative products can effectively protect the environment and save a lot of resources, so it is important for us to study the remanufacturing of innovative products. In the face of the growing market for innovative products, we can not only use the goodwill model and the classic demand model; the diffusion of innovations model to study the remanufacturing supply chain is also particularly important. Therefore, this paper introduces the diffusion of innovations model to characterize the market's changes in demand for new products. The diffusion of innovations model (Bass model) was proposed by Frank Bass [27] in 1969 to describe how durable new products expand the market among consumers. Subsequently, Bass et al. [28] proposed a diffusion model of the duopoly market under competitive conditions and studied the advertising strategy of oligopoly. Since the application of this model was quite successful, it has been widely used in various fields. Quan et al. [29] introduced the Bass model into the open-loop supply chain to characterize the immediate market demand. They studied the master–slave dynamic game process led by a supplier and followed by a retailer.

Based on the above literature, we find the following deficiencies in the existing remanufacturing supply chain studies.

1.  In the growing market of innovative products such as mobile phones, computers, automobiles, and other smart devices, scholars only consider the goodwill model and the classical demand model. Few scholars have studied the supply chain of innovative products, and the research on the supply chain of innovative products needs to be deepened.
2.  The existing research on the remanufacturing supply chain mainly focuses on the static supply chain, without considering the game between manufacturers and retailers in continuous time. The static model cannot adapt to the continuous and real-time modeling requirements, so it is urgent to increase research on the dynamic supply chain.

Therefore, this paper introduces the Bass model to establish a remanufacturing supply chain model based on the above research. Retailers are responsible for product sales and recycling, and manufacturers are responsible for new product manufacturing and recycling product remanufacturing, to find the best decision path for the best retail price, recycling effort, and wholesale price. The main contributions of this thesis are (1) the introduction of

the Bass model into the remanufacturing supply chain; (2) exploring the optimal dynamic decision path of manufacturers and retailers in the process of new product diffusion; (3) analyzing the overall long-term profits of manufacturers, retailers, and the entire supply chain under different market conditions.

The structure of this paper is as follows: Section 2 is Model Background Description, which introduces the Bass model, model-related parameters, and the remanufacturing supply chain model, respectively. In Section 3, Model Building and Solving, we solve the differential game model of remanufacturing supply chain and obtain the optimal pricing strategy for each member of the supply chain. Section 4 is optimal decision analysis, and Section 5 is the numerical example, where we analyze the impact of different innovation coefficients and imitation coefficients on the supply chain. Finally, Section 6 is conclusions.

## 2. Model Background Description

### 2.1. Introduction to the Bass Model

The core idea of the Bass model is that the group initially purchasing a new durable product is an innovative group, and their purchasing decisions are independent of other members of the social system. The group that buys the durable product later is used as an imitation group. The time for this group to buy a new product is affected by the innovation group, and this influence increases with the increase in the number of purchasers. In the context of achieving the "carbon peaking and carbon neutrality goals", customers who are the first to use remanufactured products can be regarded as innovative groups, and the other group can be regarded as imitating groups. The mathematical source of this theory is the reliability function model, which is widely used in reliability theory. The total market demand is equal to the "total number of experiments" in the reliability function. The number of products sold at time $t$ is equivalent to the "accumulated number of failed products" in the reliability function, and the number of products that have not been sold at time $t$. is equivalent to the "number of products still available" in the reliability function. The new product market share $x(t)$ at time $t$ is equivalent to the probability distribution function $F(\cdot)$ in the reliability function. Therefore, the Bass model can well describe the dynamic game evolution process of the remanufacturing supply chain of new durable products.

When a manufacturer develops a new durable product and puts it on the market, the retailer is responsible for selling the latest product and recycling it. The manufacturer is responsible for producing new products and remanufacturing waste products. The product will be replaced by next-generation products after being sold and will no longer be sold after exiting the market. As the leader of the remanufacturing supply chain, the manufacturer first determines the product's wholesale price. Then, the retailer determines its retail price and recycling strategy based on the wholesale price selected by the manufacturer.

### 2.2. Symbol Description

(1)    Main parameters and their meanings

$c_m$: the marginal production cost of new products; $c_r$: marginal production cost of remanufactured products; $\Delta$: the difference between the marginal production cost of new products and the marginal production cost of reproducts, namely $c_m - c_r$; $N$: the largest demand in the market; $a$: the influence coefficient of price on the conversion of innovators (first-time consumers) to imitators (potential consumers); $j$: innovation coefficient, which indicates the proportion of new products that can be sold without any sales effort after being launched; $k$: imitation coefficient, which indicates the influence coefficient of innovators affected by sales efforts on imitators; $\gamma$: the manufacturer's incentive coefficient for the retailer's recycling; $k_r$: the retailer's product recycling activity cost coefficient; $\alpha$: the recycling effort's influence coefficient on the recycling rate; $\beta$: the natural recession coefficient of the recovery rate; $T$: the total time the new product is put on the market; $x(t)$: the market share of the new product at time $t$; $D(t)$: the demand for new products at time $t$; $\tau(t)$: retailer's product recovery rate at time $t$.

(2) Variables (including objective function) and their meanings

$w(t)$: the wholesale price of the new product at time $t$, which represents the manufacturer's decision variable; $p(t)$: the retail price of the new product at time $t$, which represents the retailer's decision variable; $R(t)$: the retailer's recycling effort at time $t$, which represents the retailer's decision variable; $\pi_M$: the manufacturer's overall long-term profit; $\pi_R$: the overall long-term profit of the retailer; $\pi_h$: the overall long-term profit of the supply chain, namely $\pi_M + \pi_R$.

### 2.3. Optimal Control Theory

Optimal control theory is a branch of mathematical optimization that aims to find an optimal control of a dynamical system over a specific period of time, which can lead to a maximum (or minimum) value of a specific system performance index. The basic optimal control problem is described as follows:

$$\begin{aligned}
&\max V = \int_0^T F(t, y, u)dt \\
&s.t. \\
&\quad \dot{y}(t) = f(t, y, u) \\
&\quad y(0) = A \text{ , } y(T) \text{ free } (A, T \text{ given}) \\
&\quad u(\mathrm{T}) \in U \text{ for all } t \in [0, T]
\end{aligned} \tag{1}$$

where $V$ is the performance metric to be maximized, $F$ is the immediate performance metric, $t$ is the time variable, $y$ is the state variable, and $u$ is the control variable to find an optimal control path $u^*$ that maximizes the performance metric in time $[0, T]$. Another class of variables, costate variables $\lambda(t)$, is introduced in the solution process. Costate variables enter the optimal control problem through Hamilton functions. Hamilton functions are defined as follows:

$$H(t, y, u, \lambda) \equiv F(t, y, u) + \lambda(t)f(t, y, u) \tag{2}$$

The most important problem for solving optimal control is the first-order condition, also known as the maximum principle. After defining the Hamilton function, the maximum principle conditions are as follows:

$$\begin{aligned}
&\underset{u}{Max} H(t, y, u, \lambda) \text{ for all } t \in [0, T] \\
&\dot{y}(t) = \frac{\partial H}{\partial \lambda} \quad \text{(equation of motion for } y) \\
&\dot{\lambda}(T) = -\frac{\partial H}{\partial y} \quad \text{(equation of motion for } \lambda) \\
&\lambda(T) = 0 \quad \text{(transversality condition)}
\end{aligned} \tag{3}$$

Symbol $\underset{u}{\max} H$ indicates that the Hamilton function is maximized, and this equivalent condition is expressed as

$$H(t, y, u^*, \lambda) \geq H(t, y, u, \lambda) \text{ for all } t \in [0, T] \tag{4}$$

where $u^*$ is the optimal control variable. If $\underset{u}{Max} H(t, y, u, \lambda)$ is differentiable with respect to $u$, it can be represented by the first-order condition $\frac{\partial H}{\partial u} = 0$. The optimal control variables, state variables, and costate variables of the optimal control problem can be solved by the maximum principle.

### 2.4. Model Function

(1) Consumer market

An innovative product is a product with product innovation. Product innovation is defined as the development of new products, changes in design of established products, or use of new materials or components in the manufacture of established products [30], e.g., smart phones (iPhones, Samsung smart phones, etc.), smartwatches (apple watch, etc.), computers (Dell, HP, Lenovo, etc.), graphics cards (Geforce RTX graphics cards, Radeon

RX graphics cards, etc.), etc. Once sold, the innovative product will spread in the consumer market, gradually accumulating sales through publicity, promotion, and verbal communication among consumers. Assuming that the target market demand for the innovative product is $N$, according to the assumptions of the Bass model, we divide consumers into two categories. One category is consumers who have accepted and purchased new products, also known as innovators (first-time consumers). Its market share at time $t$ is $x(t)$, then the purchase quantity of the innovator at time $t$ is $Nx(t)$; The other category is consumers who have not yet accepted—but may be influenced by—innovators in the future and will accept new products in the future, also known as imitators (potential consumers). Its proportion at time $t$ is $1 - x(t)$, and the potential consumption amounts of potential consumers at time $t$ is $N(1 - x(t))$. According to Reference [29], we also assume that the change in market share is

$$\dot{x}(t) = e^{-ap(t)}(j + kx(t))(1 - x(t)) \tag{5}$$

where $e^{-ap(t)}$ represents the influence coefficient of price on the conversion of imitators to innovators. The higher the price, the slower the conversion. $kx(t)$ represents the degree of influence that innovators affected by sales efforts have on potential consumers. $j$ represents the degree of influence that innovators who are not affected by sales efforts have on potential consumers. Suppose $x(0) = 0$ indicates that the new product has just entered the market and has not yet spread among consumers. Assuming that the innovator only buys one copy of the product and does not repeat the purchase, the demand at time $t$ is equal to the amount of change in existing consumers at time $t$, namely:

$$D(t) = N\dot{x} = Ne^{-ap(t)}(j + kx(t))(1 - x(t)) \tag{6}$$

(2)  Manufacturer

As the remanufacturing supply chain leader, the manufacturer is responsible for the production and remanufacturing of new products and rewards retailers for product recycling. Assuming that there is no essential difference between a newly produced product and a remanufactured product, because through the remanufacturing of these used products, their quality and performance are brought back to the standards and requirements of new products. The marginal production cost of a manufacturer's new product is $c_m$, and the marginal production cost of a remanufactured product is $c_r$. Let $\Delta = c_m - c_r$. The average cost of producing a marginal new product is $c_m - \Delta\tau(t)$. The manufacturer sells the new product to the retailer at the wholesale price of $w(t)$, and provides the retailer with a reward of $\gamma\tau(t)$ according to the recycling ratio, where $\gamma$ is the reward coefficient and $\tau(t)$ is the recycling rate at $t$ time. Electronic durable goods such as mobile phones, TVs, and computers are updated very quickly. Manufacturers often let these products withdraw from the market after a period of time and introduce new products at the same time. Mark the total time of the new product on the market as $T$. Manufacturers pursue the maximization of their own interests by choosing the optimal path of the optimal wholesale price $w(t)$.

(3)  Retailer

As a follower of a remanufacturing supply chain, retailers are responsible for the sales of new products and recycling the products. The retailer sells new products to consumers at time $t$ at price $p(t)$. The recycling effort of the retailer at time $t$ is $R(t)$, the recycling rate of waste products at time $t$ is $\tau(t)$, and $\tau(0) = 0$ when the new product just enters the market. Assuming that the recovery effort has a positive effect on the change in the recovery rate, and considering the natural decline of the recovery rate, the change in the recovery rate is:

$$\dot{\tau}(t) = \alpha R(t) - \beta\tau(t) \tag{7}$$

where $\alpha > 0$ is the influence coefficient of the recovery effort on the change of the recovery rate, and $\beta$ is the recovery rate decline coefficient. Assuming that the marginal return of recovery costs decreases with the increase in input, the recovery effort is assumed to be a quadratic function $\frac{1}{2}k_r R^2(t)$, $k_r > 0$. After the manufacturer chooses the optimal wholesale price $w(t)$, the retailer pursues the maximization of its own interests by choosing the corresponding optimal retail price $p(t)$ and the optimal recycling effort $R(t)$.

## 3. Model Building and Solving

In this remanufacturing supply chain, both the manufacturer and the retailer constitute a Stackelberg game in which the manufacturer leads the retailer to follow. The specific design is shown in Figure 1 below. The manufacturer first decides its wholesale price $w(t)$, and then the retailer decides the retail price $p(t)$ and the recycling effort $R(t)$. The objective functions of the manufacturer and retailer can be obtained as:

$$\pi_M = \int_0^T \{[w(t) - c_m + \Delta\tau(t)]N\dot{x}(t) - \gamma\tau(t)\}dt \tag{8}$$

$$\pi_R = \int_0^T \{[p(t) - w(t)]N\dot{x}(t) - \frac{1}{2}k_r R^2(t) + \gamma\tau(t)\}dt \tag{9}$$

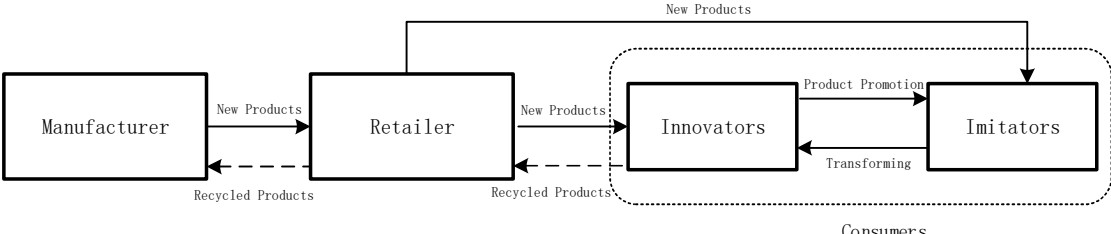

**Figure 1.** Supply chain design diagram.

When the manufacturer determines the optimal wholesale price $w(t)$, the retailer's optimal retail price $p(t)$ and the optimal recovery effort $R(t)$ are sought to maximize its profit $\pi_R$. That is, to solve the following optimal control problem:

$$max\ \pi_R = \max_{p,R} \int_0^T \{[p(t) - w(t)]N\dot{x}(t) - \frac{1}{2}k_r R^2(t) + \gamma\tau(t)\}dt \tag{10}$$

$$s.t. \begin{cases} \dot{x}(t) = e^{-ap(t)}(j + kx(t))(1 - x(t)) \\ \dot{\tau}(t) = \alpha R(t) - \beta\tau(t) \\ x(0) = 0 \\ \tau(0) = 0 \end{cases} \tag{11}$$

**Proposition 1.** *When the manufacturer's optimal wholesale price is $w^*(t)$, the retailer's optimal retail price $p^*(t)$ is:*

$$p^*(t) = \frac{1}{a} + w(t) - \frac{1}{a}ln\frac{(j + kx(T))(1 - x(T))}{(j + kx(t))(1 - x(t))} \tag{12}$$

The retailer's optimal recycling effort $R^*(t)$ is:

$$R^*(t) = \frac{\alpha\gamma}{\beta k_r(1 - e^{\beta(T-t)})} \tag{13}$$

**Proof.** According to [31]. The Hamilton function of the system is:

$$H_R = [Np(t) - Nw(t) + \lambda_{R1}(t)]e^{-ap(t)}[j + kx(t)](1 - x(t))$$
$$- \frac{1}{2}k_r R^2(t) + \gamma\tau(t) + \lambda_{R2}(t)[\alpha R(t) - \beta\tau(t)] \tag{14}$$

Necessary conditions for the retailer's optimal control problem are:

$$\begin{cases} \frac{\partial H_R}{\partial p} = 0 \\ \frac{\partial H_R}{\partial R} = 0 \\ \dot{\lambda}_{R1}(t) = -\frac{\partial H_R}{\partial x} \\ \dot{\lambda}_{R2}(t) = -\frac{\partial H_R}{\partial \tau} \\ \dot{x}(t) = e^{-ap(t)}(j + kx(t))(1 - x(t)) \\ \dot{\tau}(t) = \alpha R(t) - \beta\tau(t) \end{cases} \tag{15}$$

Transverse conditions are: $\lambda_{R1}(T) = 0, \lambda_{R2}(T) = 0, x(0) = 0, \tau(0) = 0$.

From $\frac{\partial H_R}{\partial p} = (N - aNp(t) + aNw(t) - a\lambda_{R1}(t))e^{-ap(t)}(j + kx(t))(1 - x(t)) = 0$, we can obtain

$$p(t) = \frac{1}{a} + w(t) - \frac{\lambda_{R1}}{N} \tag{16}$$

From

$$\begin{aligned} \dot{\lambda}_{R1}(t) &= -\frac{\partial H_R}{\partial x} = -(Np(t) - Nw(t) + \lambda_{R1}(t))e^{-ap(t)}(k - j - 2kx(t)) \\ &= -(Np(t) - Nw(t) + \lambda_{R1}(t))\dot{x}(t)\frac{(k-j-2kx(t))}{(j+kx(t))(1-x(t))} \\ &= -\dot{x}(t)\frac{N(k-j-2kx(t))}{a(j+kx(t))(1-x(t))} \end{aligned} \tag{17}$$

then

$$\int_t^T \dot{\lambda}_{R1}(t)dt = -\int_t^T \dot{x}(t)\frac{N(k - j - 2kx(t))}{a(j + kx(t))(1 - x(t))}dt$$

$$\lambda_{R1}(T) - \lambda_{R1}(t) = -\int_{x(t)}^{x(T)} \frac{N(k - j - 2kx(t))}{a(j + kx(t))(1 - x(t))}dx \tag{18}$$

$$\lambda_{R1}(t) = \frac{N}{a}\ln\frac{(j + kx(T))(1 - x(T))}{(j + kx(t))(1 - x(t))}$$

Bring it into Equation (12) to obtain

$$p(t) = \frac{1}{a} + w(t) - \frac{1}{a}\ln\frac{(j + kx(T))(1 - x(T))}{(j + kx(t))(1 - x(t))} \tag{19}$$

From $\frac{\partial H_R}{\partial R} = -k_r R(t) + \alpha\lambda_{R2}(t) = 0, \dot{\lambda}_{R2}(t) = -\frac{\partial H_R}{\partial \tau} = \beta\lambda_{R2}(t) - \gamma$, we can obtain

$$R(t) = \frac{\alpha\lambda_{R2}(t)}{k_r} \tag{20}$$

$$\lambda_{R2}(t) = C_1 e^{\beta t} + \frac{\gamma}{\beta} \tag{21}$$

From $\lambda_{R2}(T) = 0$, we obtain $C_1 = -\frac{\gamma}{\beta}e^{-\beta T}$. Putting it into Equation (21), we obtain $\lambda_{R2}(t) = \frac{\gamma}{\beta}(1 - e^{\beta(t-T)})$, and then

$$R^*(t) = \frac{\alpha\gamma}{\beta k_r}(1 - e^{\beta(t-T)}) \tag{22}$$

Substituting Equation (22) into $\dot{\tau}(t) = \alpha R(t) - \beta\tau(t)$, and $\tau(0) = 0$, we obtain

$$\tau^*(t) = \frac{\alpha^2\gamma}{\beta^2 k_r}\left(1 - e^{-\beta t} + \frac{1}{2}e^{-\beta T}(e^{-\beta t} - e^{\beta t})\right) \tag{23}$$

The proof is complete. □

**Proposition 2.** *The manufacturer's optimal retail price* $w^*(t)$ *is*

$$w^*(t) = \frac{1}{a} + c_m - \Delta\tau(t) = \frac{1}{a} + c_m - \Delta(\frac{\alpha^2\gamma}{\beta^2 k_r}(1 - e^{-\beta t} + \frac{1}{2}e^{-\beta T}(e^{-\beta t} - e^{\beta t}))) \quad (24)$$

**Proof.** The Hamilton function of the system is

$$H_M = (Nw(t) - Nc_m + N\Delta\tau(t) + \lambda_M(t))e^{-(1+aw(t))}(j + kx(T))(1 - x(T)) - \gamma\tau(t) \quad (25)$$

The necessary conditions for the manufacturer's optimal control problem are

$$\begin{cases} \frac{\partial H_M}{\partial w} = 0 \\ \dot{\lambda}_M(t) = -\frac{\partial H_M}{\partial x} \\ \dot{x}(t) = e^{-(1+aw(t))}(j + kx(T))(1 - x(T)) \end{cases} \quad (26)$$

The transverse conditions are $\lambda_M(T) = 0$, $x(0) = 0$.
From

$$\frac{\partial H_M}{\partial w} = (N - aNw(t) + aNc_m - aN\Delta\tau(t) - a\lambda_M(t))e^{-(1+aw(t))}(j + kx(T))(1 - x(T)) = 0 \quad (27)$$

we can obtain

$$w(t) = \frac{1}{a} + c_m - \Delta\tau(t) - \frac{\lambda_M(t)}{N} \quad (28)$$

and $\dot{\lambda}_M(t) = -\frac{\partial H_M}{\partial x} = 0$, $\lambda_M(T) = 0$, so $\lambda_M(t) \equiv 0$, then

$$w^*(t) = \frac{1}{a} + c_m - \Delta\tau(t) = \frac{1}{a} + c_m - \Delta\frac{\alpha^2\gamma}{\beta^2 k_r}(1 - e^{-\beta t} + \frac{1}{2}e^{-\beta T}(e^{-\beta t} - e^{\beta t})) \quad (29)$$

The proof is complete. □

**Proposition 3.** *For the above system, the optimal paths for manufacturers and retailers in a remanufacturing supply chain dominated by manufacturers are:*

(1) *The manufacturer's optimal pricing path strategy is*

$$w^*(t) = \frac{1}{a} + c_m - \Delta(\frac{\alpha^2\gamma}{\beta^2 k_r}(1 - e^{-\beta t} + \frac{1}{2}e^{-\beta T}(e^{-\beta t} - e^{\beta t}))) \quad (30)$$

(2) *The retailer's optimal pricing path strategy and recycling effort decision are*

$$p^*(t) = \frac{1}{a} + w(t) - \frac{1}{a}ln\frac{(j + kx(T))(1 - x(T))}{(j + kx(t))(1 - x(t))} \quad (31)$$

$$R^*(t) = \frac{\alpha\gamma}{\beta k_r}(1 - e^{\beta(t-T)}) \quad (32)$$

See Propositions 1 and 2 for specific proofs.

**Proposition 4.** *The path of new product market share is*

$$x(t) = B(t)(j + kx(T))(1 - x(T)) \quad (33)$$

*where*

$$B(t) = \int_0^t e^{-(2+ac_m - a\Delta(\frac{\alpha^2\gamma}{\beta^2 k_r}(1-e^{-\beta t}+\frac{1}{2}e^{-\beta T}(e^{-\beta t}-e^{\beta t}))))} dt \tag{34}$$

*and*

$$x(T) = \frac{-1 - B(T)j + B(T)k + \sqrt{4B^2(T)jk + (-1-B(T)j+B(T)k)^2}}{2B(T)k} \tag{35}$$

**Proof.** From Proposition 2:

$$\begin{aligned} x(t) &= (j+kx(T))(1-x(T))\int_0^t e^{-(1+aw(t))}dt \\ &= (j+kx(T))(1-x(T))\int_0^t e^{-(2+ac_m-a\Delta(\frac{\alpha^2\gamma}{\beta^2 k_r}(1-e^{-\beta t}+\frac{1}{2}e^{-\beta T}(e^{-\beta t}-e^{\beta t}))))} dt \end{aligned} \tag{36}$$

Let $B(t) = \int_0^t e^{-(2+ac_m-a\Delta(\frac{\alpha^2\gamma}{\beta^2 k_r}(1-e^{-\beta t}+\frac{1}{2}e^{-\beta T}(e^{-\beta t}-e^{\beta t}))))} dt$, then

$$x(T) = \frac{-1 - B(T)j + B(T)k + \sqrt{4B^2(T)jk + (-1-B(T)j+B(T)k)^2}}{2B(T)k}$$

When $t = T$, $x(T) = B(t)(j + kx(T))(1 - x(T))$, then

$$x(T) = \frac{-1 - B(T)j + B(T)k + \sqrt{4B^2(T)jk + (-1-B(T)j+B(T)k)^2}}{2B(T)k},$$

or $x(T) = \frac{-1-B(T)j+B(T)k-\sqrt{4B^2(T)jk+(-1-B(T)j+B(T)k)^2}}{2B(T)k} < 0$ (drop out).
    The proof is complete. $\square$

**Proposition 5.** *The long-term profit of the manufacturer is*

$$\pi_M = \frac{N}{a}x(T) - \frac{\alpha^2\gamma^2}{2\beta^3 k_r}(2\beta T - 3 + 4e^{-\beta T} - e^{-2\beta T}) \tag{37}$$

*The long-term benefits of the retailer are*

$$\pi_R = \frac{N}{a}\left(x(T) + ln\frac{(j+kx(T))(1-x(T))}{j}\right) + \frac{\alpha^2\gamma^2}{4\beta^3 k_r}(2\beta T - 3 + 4e^{-\beta T} - e^{-2\beta T}) \tag{38}$$

**Proof.** The long-term benefits of the manufacturer are

$$\begin{aligned} \pi_M &= \int_0^T (w(t) - c_m + \Delta\tau(t))N\dot{x}(t) - \gamma\tau(t)dt \\ &= \int_0^T \frac{N}{a}\dot{x}(t)dt - \int_0^T \gamma\tau(t)dt \\ &= \frac{N}{a}x(T) - \frac{\alpha^2\gamma^2}{2\beta^3 k_r}(2\beta T - 3 + 4e^{-\beta T} - e^{-2\beta T}) \end{aligned} \tag{39}$$

The long-term benefits of the retailer are

$$\begin{aligned} \pi_R &= \int_0^T (p(t) - w(t))N\dot{x}(t) - \frac{1}{2}k_r R^2(t) + \gamma\tau(t)dt \\ &= \int_0^T (\frac{1}{a} - \frac{1}{a}ln\frac{(j+kx(T))(1-x(T))}{(j+kx(t))(1-x(t))})N\dot{x}(t) - \frac{1}{2}k_r R^2(t) + \gamma\tau(t)dt \\ &= \frac{1}{a}\int_{x(0)}^{x(T)} (1 - ln\frac{(j+kx(T))(1-x(T))}{(j+kx(t))(1-x(t))})Ndx - \int_0^T \frac{\alpha^2\gamma^2}{2\beta^2 k_r}(1-e^{\beta(t-T)})^2 dt + \int_0^T \gamma\tau(t)dt \\ &= \frac{1}{a}\int_{x(0)}^{x(T)} (1 - ln\frac{(j+kx(T))(1-x(T))}{(j+kx(t))(1-x(t))})Ndx + \frac{\alpha^2\gamma^2}{4\beta^3 k_r}(2\beta T - 3 + 4e^{-\beta T} - e^{-2\beta T}) \\ &= \frac{N}{a}\left(x(T) - \int_{x(0)}^{x(T)} ln\frac{(j+kx(T))(1-x(T))}{(j+kx(t))(1-x(t))}dx\right) + \frac{\alpha^2\gamma^2}{4\beta^3 k_r}(2\beta T - 3 + 4e^{-\beta T} - e^{-2\beta T}) \end{aligned} \tag{40}$$

The proof is complete. □

## 4. Optimal Decision Analysis

**Result 1.** *Although the manufacturer's optimal wholesale price changes over time, the marginal contribution of each new product is $\frac{1}{a}$. That is to say, the optimal wholesale price strategy for the manufacturer to obtain the maximum profit must make the marginal contribution of each new product $\frac{1}{a}$.*

**Result 2.** *According to Equation (31), $\frac{\partial(p-w)}{\partial x} = \frac{k-j-2kx(t)}{a(j+kx(t))(1-x(t))}$ is obtained. When $k-j>0$, that is, when the influence of the innovator on the purchase of imitators is greater than the influence of the imitator's innovative purchase effect, the retailer sets prices so that the marginal contribution of each product increases first and then decreases as the proportion of innovators increases; when $k-j \leq 0$, that is, when the influence of the imitator's innovative purchase effect is greater than the influence of the innovator on the imitator's purchase, the retailer's pricing makes the marginal contribution of each product decrease as the proportion of adopters increases.*

**Result 3.** *From Equation (32), we can see that $\frac{dR(t)}{dt} = -\frac{\alpha\gamma}{k_r}e^{\beta(t-T)} < 0$. The recycling effort of retailers decreases with time. The recycling effort is greatest when the new product is first launched. With the continuation of the launch time, the recycling effort gradually decreases. When the product is withdrawn from the market, the recycling effort is reduced to 0.*

**Result 4.** *From Equation (35), we know that $\frac{\partial x(T)}{\partial j} = \frac{B(T)(1+B(T)j+B(t)k-\sqrt{(1+B(T)j+B(t)k)^2-4B(T)j})}{2B(T)k\sqrt{4B^2(T)jk+(-1-B(T)j+B(T)k)^2}} > 0$. That is, the greater the innovation coefficient of a product, the greater the final sales volume $Nx(T)$ of the product, and it can be seen from Equation (37) that the overall long-term profit of the manufacturer will also be greater.*

**Result 5.** *From Equation (35), we can see that $\frac{\partial x(T)}{\partial T} = \frac{B'(T)[\sqrt{4B^2(T)jk+(-1-B(T)j+B(T)k)^2}-(1+B(T)j-B(T)k)]}{2kB^2(T)\sqrt{4B^2(T)jk+(-1-B(T)j+B(T)k)^2}}$. From $B'(T)>0$, we know that $\frac{\partial x(T)}{\partial T} > 0$. That is, the longer the selected sales period, the higher the market share $x(T)$ when the product exits the market, and the greater the final sales volume $Nx(T)$ will be.*

## 5. Numerical Example

The iPhone is a typical representative of innovative products; according to TechInsights' teardown analysis, the estimated cost of the 256 GB version of the iPhone 13 Pro is around USD 500, and its sell price is USD 1099, i.e., $c_m = 500$. Apple Inc., Cupertino, CA, USA, which uses recycling technology to disassemble and recycle old iPhones, has set up a special department to study how to improve the technology of recycling old iPhones. They have developed three devices, named Daisy, Taz, and Dave, to take apart old iPhones and collect the valuable raw materials for making new products. Nearly 20% of Apple products shipped in fiscal year 2021 were made from recycled materials [32], so we can obtain $c_r = 400$, $\Delta = 100$. Based on the above references [20–25,29], we can obtain $a = 0.002$, $k_r = 2000$, $\gamma = 10000$, $\alpha = 0.2$, $\beta = 0.5$, $T = 30$, $N = 100,000$. The numerical analysis is performed by bringing these values into the model as follows.

The overall long-term profit of the manufacturer, the overall long-term profit of the retailer, the overall long-term profit of the supply chain, and the proportion of innovators at time $T$ vary with $j$ and $k$, as shown in the figures below.

**Result 6.** Table 1 and Figures 2–5 show that when the innovation coefficient $j$ and the imitation coefficient $k$ are larger, the proportion of innovators at time $T$ is larger, and the overall long-term profit of the manufacturer and retailer is higher.

**Table 1.** The influence of changes in parameters *j* and *k* on $X(T)$, $\pi_m$, $\pi_r$, and $\pi_h$.

| | *k* = 0.1 | *k* = 0.3 | *k* = 0.5 | *k* = 0.7 | *k* = 0.9 |
|---|---|---|---|---|---|
| *j* = 0.1 | $x(T)$ = 16.77% <br> $\pi_m$ = 8,169,477 <br> $\pi_r$ = 8,653,421 <br> $\pi_h$ = 16,822,898 | $x(T)$ = 22.39% <br> $\pi_m$ = 10,978,126 <br> $\pi_r$ = 10,149,939 <br> $\pi_h$ = 21,128,065 | $x(T)$ = 30.24% <br> $\pi_m$ = 14,903,680 <br> $\pi_r$ = 12,203,997 <br> $\pi_h$ = 27,103,677 | $x(T)$ = 39.29% <br> $\pi_m$ = 19,428,616 <br> $\pi_r$ = 14,857,422 <br> $\pi_h$ = 34,286,038 | $x(T)$ = 47.76% <br> $\pi_m$ = 23,665,577 <br> $\pi_r$ = 17,959,185 <br> $\pi_h$ = 41,624,762 |
| *j* = 0.3 | $x(T)$ = 36.75% <br> $\pi_m$ = 18,160,691 <br> $\pi_r$ = 21,972,993 <br> $\pi_h$ = 40,133,684 | $x(T)$ = 42.44% <br> $\pi_m$ = 21,005,584 <br> $\pi_r$ = 24,194,929 <br> $\pi_h$ = 45,200,513 | $x(T)$ = 48.31% <br> $\pi_m$ = 23,937,359 <br> $\pi_r$ = 26,665,438 <br> $\pi_h$ = 50,602,797 | $x(T)$ = 53.89% <br> $\pi_m$ = 26,727,890 <br> $\pi_r$ = 29,315,981 <br> $\pi_h$ = 56,043,871 | $x(T)$ = 58.89% <br> $\pi_m$ = 29,226,630 <br> $\pi_r$ = 32,065,608 <br> $\pi_h$ = 61,292,238 |
| *j* = 0.5 | $x(T)$ = 48.63% <br> $\pi_m$ = 24,099,991 <br> $\pi_r$ = 32,304,674 <br> $\pi_h$ = 56,404,665 | $x(T)$ = 53.24% <br> $\pi_m$ = 26,402,195 <br> $\pi_r$ = 34,592,296 <br> $\pi_h$ = 60,994,491 | $x(T)$ = 57.63% <br> $\pi_m$ = 28,597,894 <br> $\pi_r$ = 36,981,292 <br> $\pi_h$ = 65,579,186 | $x(T)$ = 61.65% <br> $\pi_m$ = 30,608,553 <br> $\pi_r$ = 39,425,744 <br> $\pi_h$ = 70,034,297 | $x(T)$ = 65.23% <br> $\pi_m$ = 32,397,806 <br> $\pi_r$ = 41,884,117 <br> $\pi_h$ = 74,281,923 |
| *j* = 0.7 | $x(T)$ = 56.63% <br> $\pi_m$ = 28,098,253 <br> $\pi_r$ = 40,789,355 <br> $\pi_h$ = 68,889,608 | $x(T)$ = 60.32% <br> $\pi_m$ = 29,944,437 <br> $\pi_r$ = 429,90,543 <br> $\pi_h$ = 72,934,980 | $x(T)$ = 63.74% <br> $\pi_m$ = 31,654,302 <br> $\pi_r$ = 45,225,646 <br> $\pi_h$ = 76,879,948 | $x(T)$ = 66.84% <br> $\pi_m$ = 33,201,773 <br> $\pi_r$ = 47,466,725 <br> $\pi_h$ = 80,668,498 | $x(T)$ = 69.59% <br> $\pi_m$ = 34,580,783 <br> $\pi_r$ = 49,690,895 <br> $\pi_h$ = 84,271,678 |
| *j* = 0.9 | $x(T)$ = 62.42% <br> $\pi_m$ = 30,992,496 <br> $\pi_r$ = 48,004,289 <br> $\pi_h$ = 78,996,785 | $x(T)$ = 65.42% <br> $\pi_m$ = 32,493,335 <br> $\pi_r$ = 50,080,667 <br> $\pi_h$ = 82,574,002 | $x(T)$ = 68.17% <br> $\pi_m$ = 33,866,554 <br> $\pi_r$ = 52,158,179 <br> $\pi_h$ = 86,024,733 | $x(T)$ = 70.64% <br> $\pi_m$ = 35,105,751 <br> $\pi_r$ = 54,219,755 <br> $\pi_h$ = 89,325,506 | $x(T)$ = 72.86% <br> $\pi_m$ = 36,214,147 <br> $\pi_r$ = 56,252,149 <br> $\pi_h$ = 92,466,296 |

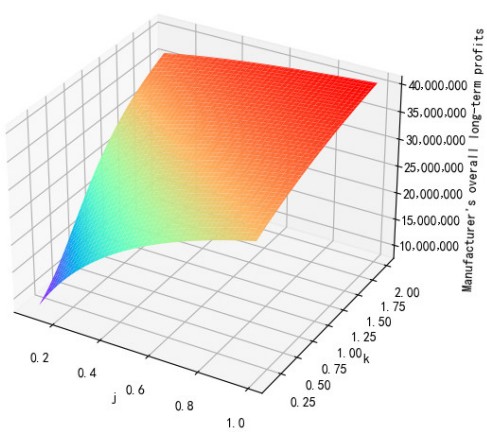

**Figure 2.** The relationship between parameters *j*, *k* and the overall long-term profit of the manufacturer.

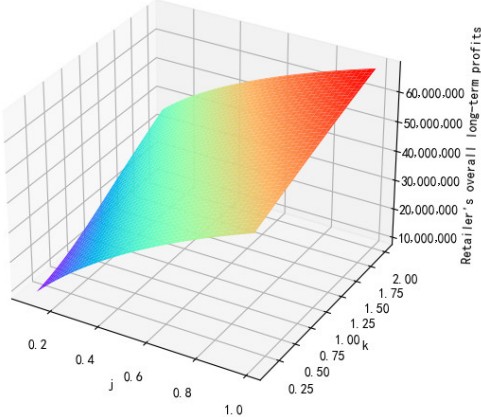

**Figure 3.** The relationship between parameters *j*, *k* and the overall long-term profit of the retailer.

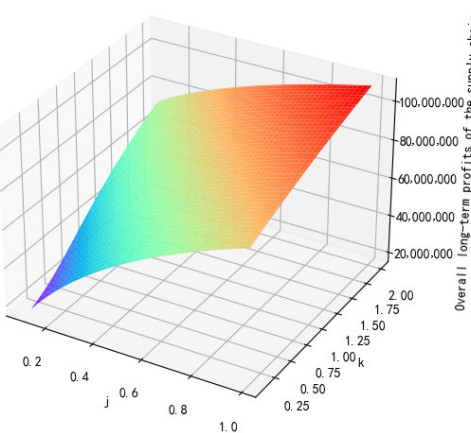

**Figure 4.** The relationship between parameters $j$, $k$ and the overall long-term profit of the supply chain.

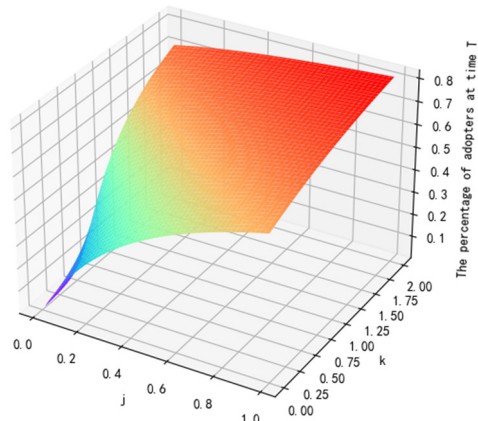

**Figure 5.** The relationship between parameters $j$, $k$ and the proportion of innovators at time $T$.

Observing Figures 2 and 4, we can find that when the innovation coefficient $j$ and the imitation coefficient $k$ change, the overall long-term profit of the manufacturer is similar to the change graph of the proportion of innovators at time $T$. Because the amount of change in $\frac{\alpha^2\gamma^2}{2\beta^3 k_r}(2\beta T - 3 + 4e^{-\beta T} - e^{-2\beta T})$ in $\pi_M = \frac{N}{a}x(T) - \frac{\alpha^2\gamma^2}{2\beta^3 k_r}(2\beta T - 3 + 4e^{-\beta T} - e^{-2\beta T})$ is much smaller than the amount of change in $\frac{N}{a}x(T)$, when $j$, $k$, $\alpha$, and $\beta$ change, the overall long-term profit of the manufacturer and the proportion of innovators at time $T$ are roughly linear. Similarly, the image of the retailer's overall long-term profit is similar to the image of the total supply chain profit because the degree of change in the overall long-term profit of the retailer is greater than the degree of change in the overall long-term profit of the manufacturer.

From Proposition 3, it can be seen that the parameters $j$ and $k$ do not affect the recovery effort and recovery rate. The change path of the recovery effort and recovery rate in the above situation is shown in the figure below.

**Result 7.** *Observing Figure 6, it is found that it is consistent with Result 3, and the degree of recovery effort decreases with time. When $t < 25$, it decreases slowly; when $t > 25$, it decreases sharply; when the product is withdrawn from the market ($t = 30$), the recovery intensity decreases to 0. Figure 7 shows that the recovery first gradually increases, maintains a high level of recovery from around $t = 5$ to around $t = 22$, and then begins to gradually decrease after reaching the highest recovery.*

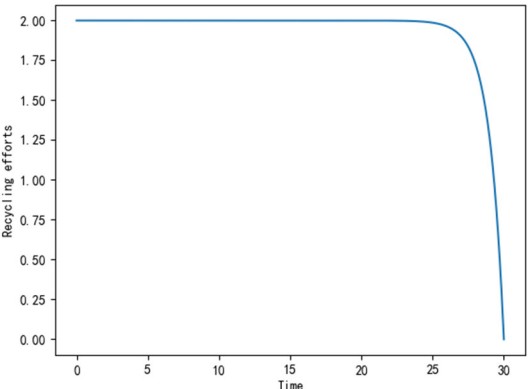

**Figure 6.** Change path of recycling effort.

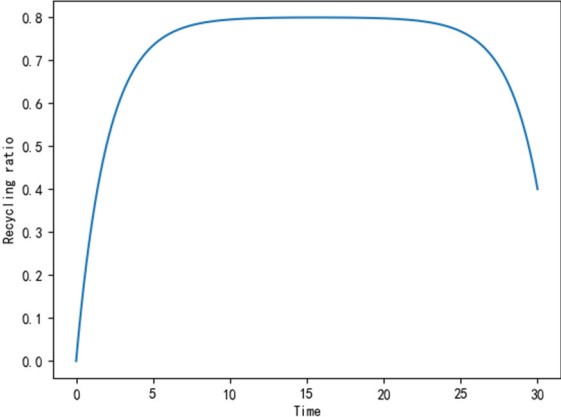

**Figure 7.** Change path of recovery rate.

## 6. Conclusions

This paper establishes a remanufacturing supply chain modeled by a single manufacturer and followed by a single retailer based on the Bass model. The optimal control method obtains the best path for the manufacturer's wholesale price, the retailer's retail price, and the best path for recycling efforts, and further obtains the optimal market share, recovery rate, and overall long-term profits of manufacturers, retailers, and supply chains. It also compares the supply chain's overall long-term profit and market share with different innovation coefficients and imitation coefficients. The main conclusions are as follows:

Conclusion (1): From Result 1, when the marginal profit of each product is $\frac{1}{a}$, the manufacturer can obtain the optimal path for the wholesale price. The retailer's pricing strategy depends on different imitation coefficients and innovation coefficients. It can be seen from Result 2 that when the imitation coefficient is greater than the innovation coefficient, retailer pricing makes the marginal profit of each product increase first and then decrease as the proportion of innovators increases. This shows that in real life, imitators limit the curiosity of new products. When a new product was first launched, the imitators became more curious. Over time, this curiosity will gradually disappear. This also reminds the participants in the supply chain that when making sales efforts (such as advertising) for a new product, it is better to do so in the early stages of product launch than in the middle and late stages. When the imitation coefficient is less than or equal to the Innovation coefficient, retailer pricing will make the marginal profit of each product decrease as the proportion of innovators increases. This shows that when more and more people use a new product in real life, the price reduction will become more and more apparent. This conclusion is consistent with objective reality. This also shows that there is no need to increase sales efforts for a new product in the middle and late stages. When the initial

innovation population (innovation coefficient *j*) is small, the product market is mainly expanded by increasing sales efforts (increasing the imitation coefficient *k*).

Conclusion (2): Results 3 and 7 show that retailers' recycling efforts maintain a high level when new products enter the market and then slowly reduce the degree of their recycling efforts. At the end of the product sales period, the attempt to recycle the product was reduced to a greater extent until the product was no longer recycled. The corresponding recovery rate gradually increased as the retailer's recovery efforts strengthened. Then, the recovery efforts of the retailer continued to weaken and then began to decline after the recovery rate reached the maximum. When many products were first launched in real life, the product recycling promotion was powerful and maintained for an extended period, and the recycling rate was also increasing. However, as the product launch time increased, the recycling effort gradually weakened, and the recycling rate reached its peak and began to decline. When the product was close to exiting the market, the promotion of product recycling would drop significantly. When the product was withdrawn from the market, the product would no longer be recycled. From Result 5, the longer the product sales time, the greater the sales volume under its optimal strategy. These conclusions are consistent with reality.

Conclusion (3): Results 4 and 6 show that in a market with a higher innovation coefficient *j* and imitation coefficient *k*, manufacturers and retailers can obtain higher profits, so the overall long-term profits of the supply chain will be higher as well. The larger the innovation coefficient, the more attractive the innovative product is to consumers; the larger imitation coefficient means the larger the customer group that imitates, thus leading to the faster diffusion of the innovative product in the market, i.e., the larger the sales quantity will be at a fixed sales time. As a result, the manufacturer and the retailer can all earn higher profits, as can the entire supply chain.

Conclusion (4): It can be seen from Result 6 that in a market with a high imitation coefficient and innovation coefficient, the proportion of innovators at the end of product sales is also relatively high. That is, more products can be sold in such a market. Fewer products are sold in markets where both the imitation coefficient and the innovation coefficient are small. The innovation coefficient is formed naturally, and the human factor is relatively small. However, the imitation coefficient can be improved by increasing sales efforts. Therefore, when the innovation coefficient is not significant, the participants in the supply chain must increase their sales efforts in the early stage of sales and strive to improve the imitation coefficient to expand sales.

Conclusion (5): Compared with Reference [29], the decision making of supply chain participants is more complicated in this paper. The manufacturer's wholesale price strategy is no longer a fixed value but changes over time. Retailers' retail prices also change with changes in wholesale prices. The proportion of new product innovators is no longer a linear function of time.

Our findings have following managerial implications:

1. To maximize profits, the manufacturer's optimal pricing strategy is to maintain a fixed value of marginal profit for each product, and in the early stage, the innovation factor is increased through advertising and other sales efforts to speed up the product diffusion, which can increase the final sales of the product.

2. The specific pricing strategy of the retailer should be determined according to different markets, and the recycling strategy of the retailer should be to make a lot of recycling publicity and preparation in the early stage to raise consumers' recycling awareness as soon as possible, so that the recycling rate of innovative products can quickly reach a high level, and then the retailer can maintain this high recycling rate with lower recycling costs.

3. In markets with higher innovation and imitation coefficients, the sales of the innovative product are higher and the profits of the closed-loop supply chain are higher, so manufacturers and retailers should choose to put the innovative product into markets with higher innovation and imitation coefficients, while in markets with lower

innovation and imitation coefficients, manufacturers and retailers should consider sales efforts such as advertising and better customer services to improve consumer satisfaction and increase the speed of product diffusion, thus increasing the profits of manufacturers and retailers.

4. The closed-loop innovative product supply chain is much more complex than the open-loop one. In the closed-loop innovative product supply chain, the dynamic demand changes in the market become more complex, and thus the optimal decisions of manufacturers and retailers are also more complex, so the members of the closed-loop supply chain need to have stronger market information acquisition ability and faster response to market changes.

The research of this paper is based on the premise of a complete market monopoly. It only considers the situation of a secondary remanufacturing supply chain when a dominant supplier and the following retailer are selling new products. And it assumes that the time when the product is no longer sold is specific. In the future, we can also consider an oligopoly situation, such as one supplier and two retailers, or two suppliers and one retailer. The time for new product sales to exit the market can be used as a control variable to find the optimal time for product replacement.

**Author Contributions:** Conceptualization, L.L. and Z.L.; methodology, Y.P.; software, N.W.; validation, L.L., Y.P. and N.W.; formal analysis, Z.L.; investigation, Z.L.; resources, L.L.; data curation, N.W.; writing—original draft preparation, Z.L.; writing—review and editing, L.L.; visualization, Y.P.; supervision, Y.P.; project administration, L.L.; funding acquisition, L.L. All authors have read and agreed to the published version of the manuscript.

**Funding:** This research was funded by National Natural Science Foundation of China (No. 72162015), Social Science Foundation of Jiangxi Province (21GL17).

**Data Availability Statement:** Not applicable.

**Acknowledgments:** This work was supported by National Natural Science Foundation of China (No. 72162015), Social Science Foundation of Jiangxi Province (21GL17). This support is gratefully appreciated.

**Conflicts of Interest:** The authors declare no conflict of interest.

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
