# Peer review of "Dynamic Optimal Decision Making of Innovative Products’ Remanufacturing Supply Chain"

_processes, doi:10.3390/pr11010295_

Round 1

Reviewer 1 Report (Previous Reviewer 1)

1. "European countries have the world’s most perfect waste information and communication equipment recycling management system"- It is dangerous to state it is a nearly perfect system. It is indeed far from a perfect system. Revise the sentence

2. Kindly revise all the typos e,g, missing full stop lines 218

3. Would be great to find examples of innovative products.

4. Design of the research methodology was not clearly mentioned.

5. Figure 2,3,4. (The numbers and axis are hard to read)

4. The discussion is a bit shallow. what are the interesting topics that can be discussed based on the results?

Author Response

Reviewer 2 Report (Previous Reviewer 3)

The references should be checked as follows;

[24] Song J, Bian Y, Liu G. Decisions of Closed-Loop Supply Chain Based on Recycling Effort and Differential Game[J]. 576 Discrete Dynamics in Nature and Society, 2020, 2020.

Author Response

This manuscript is a resubmission of an earlier submission. The following is a list of the peer review reports and author responses from that submission.

Round 1

Reviewer 1 Report

1. The white paper 2018 on intelligent remanufacturing products edited by Tsinghua University..-kindly insert the citation for the source.

2. European countries have the world’s 38 most perfect waste information and communication equipment recycling management 39 system.- a statement that need to be justified by evivdence.

3. It is good to have a definition of remanufacturing supply chain in the introduction section as it will be inform a wide range of readers.

4. Scholars at home and abroad-what does it mean scholars at home? 

5.What are the dynamic characteristics in recycling process to differentiate with the static models, that seems as the gap in the literature. List/explain them.

6. The research design was not specifically mentioned.

7. Contributions to the practitioner was not explained.

Reviewer 2 Report

The paper focuses on an important issue related to sustainability, that is, remanufacturing. However, the proposed methodology needs many improvements. Consider the following points before resubmission.

Authors need to explain the innovative products and why remanufacturing of innovative products is considered.

 The authors need to discuss the importance of the proposed methodology. Why proposed framework is important in reference to the current literature

The literature review needs to be updated. consider recent papers from 2021-2022.

I suggest implementing the developed model in real-life case examples.

Why did the authors feel that only manufacturers and retailers are important to consider in this game model?

In a real-life scenario, how kx(t) can be estimated?

Also, the model assuming that new innovative products are the same as remanufactured products from waste material seems illogical. Need more explanation to clarify this assumption.

Reviewer 3 Report

Sections 3 and 4 should be supported by the literature.
